# A location discrete choice model of crime: Police elasticity and optimal deployment

**Douglas Newball-Ramírez**[1], **Álvaro J. Riascos Villegas**[1,2]*, **Andrés Hoyos**[2], **Mateo Dulce Rubio**[3]

**1** Quantil, Bogotá, Colombia, **2** Facultad de Economia, Universidad de los Andes, Bogotá, Colombia, **3** Department of Statistics & Data Science and Heinz College of Information Systems & Public Policy, Carnegie Mellon University, Pittsburgh, PA, United States of America, **4** Departamento de Matemáticas, Facultad de Ciencias, Universidad Nacional de Colombia, Bogotá, Colombia

* ariascos@uniandes.edu.co

**Data Availability Statement:** The data underlying the results presented in the study is available from the following repository: https://github.com/lgomezt/A-Location-Discrete-Choice-Model-of-Crime/tree/main The original data from Blattman et.

## Abstract

Despite the common belief that police presence reduces crime, there is mixed evidence of such causal effects in major Latin America cities. In this work we identify the casual relationship between police presence and criminal events by using a large dataset of a randomized controlled police intervention in Bogotá D.C., Colombia. We use an Instrumental Variables approach to identify the causal effect of interest. Then we consistently estimate a Conditional Logit discrete choice model with aggregate data that allow us to identify agents' utilities for crime location using Two Stage Least Squares. The estimated parameters allow us to compute the police own and cross-elasticities of crime for each of the spatial locations and to evaluate different police patrolling strategies. The elasticity of crime to police presence is, on average across spatial locations, −0.26 for violent crime, −0.38 for property crime and −0.38 for total crime, all statistically significant. Estimates of cross-elasticities are close to zero; however, spillover effects are non-negligible. Counterfactual analysis of different police deployment strategies show, for an optimal allocating algorithm, an average reduction in violent crime of 7.09%, a reduction in property crimes of 8.48% and a reduction in total crimes of 5.15% at no additional cost. These results show the potential efficiency gains of using the model to deploy police resources in the city without increasing the total police time required.

## Introduction

Crime prediction is now ubiquitous in crime prevention and police resource planning. Studies on crime prediction and identification of risky spatial regions –the so-called "hot spots"– already exist in a vast quantity: [1–11], to name a few. However, identifying and quantitatively characterizing the causal drivers of crime is a harsh scientific problem: the optimal allocation of police resources is guided by police deployment strategies (e.g., prediction models), which at the same time determine what crime incidents are reported or how crime is displaced from one sector to another in a city. In fact, one can hardly take a stance on the causal relationship

al 2021 study is available at: https://dataverse.harvard.edu/dataset.xhtml?persistentId=doi:10.7910/DVN/XUVD77.

**Funding:** Results of the project: "Diseño y validación de modelos de analítica predictiva de fenómenos de seguridad y convivencia para la toma de decisiones en Bogotá" funded by Colciencias with resources from the Sistema General de Regalías, BPIN 2016000100036. The opinions expressed are solely those of the authors. The funders had no role in study design, data collection and analysis, decision to publish, or preparation of the manuscript.

**Competing interests:** The authors have declared that no competing interests exist.

between police presence in a particular spot and crime incidents. On the one hand, the police presence at a particular spot tends to deter crime, but at the same time crime incidents call for police presence at specific sectors of a city. Thus, it is uncertain whether police presence is a cause or a consequence of the level of crime.

Untangling this casual relationship is an instance of what [12] has called the fundamental problem of causal inference: *what would have happened had the police not been present in a particular sector of the city?* The gold standard to address this causal problem is the use of experimental data specially tailored toward this goal (i.e., a randomized controlled trial, or RCT). Running RCTs to evaluate the effect of patrols, time exposure or other intervention strategies in different regions of the city is difficult due to ethical concerns and high costs. Nevertheless, there are already several experimental or quasi-experimental studies addressing this issue, especially for US cities, but the evidence on the effect of increasing police presence in hotspots has been mixed, specially for major cities in Latin America (see [13] for a comprehensive study of proactive policing in the US).

In this study, we use [14] dataset which studies a placed-based police and city services intervention in Bogotá, Colombia (we are thankful to the authors for allowing us to use their data set for this study). [14] found that increasing police presence has modest direct impacts, even when focusing on the highest-crime hot spots, and with crime displaced nearby ruling out total reductions in crime of more than 2%. We propose a different identification strategy using an Instrumental Variables approach and a discrete choice model of spatial selection of crime to study the impact of police time exposure on crime levels in different sectors of the city. In detail, we are interested in studying the average causal effect on crime reduction of increasing police patrol time. Moreover, we are interested in estimating the own- and cross-elasticities of crime to patrol time at different locations, which captures the percentage change in crime, if any, that results from a 1% increase in the police patrol time that such location receives. These can also be interpreted as heterogeneous treatment effects for each spatial location. Finally, we use the estimated elasticities to compare counterfactual policy scenarios in search of more efficient patrolling strategies.

To summarize, our contribution is: (1) We use a unique large experimental dataset that allows for the identification of the causal effect of police patrolling on crime, (2) We estimate a spatial discrete choice model with aggregate data that allow us to identify agents' utilities, (3) Provided our estimates of the structural parameters that determine the crime location choice of the offenders, we estimate the police own- and cross-elasticity of crime for each of the quadrants (i.e., we compute where police patrolling is more effective), and (4) Based on the same structural parameters and the estimated elasticities, we evaluate different police patrolling counterfactual strategies without increasing the total police time available. In particular, we test what would have happened had the police patrol time been allocated: (a) uniformly across quadrants, (b) proportional to the incidence of crimes, (c) such that the more insecure and elastic quadrants receive a 10, 20, 30, 40, 50, 60, 70, 80, 90, and 100% increase, and (d) optimally: we solve an optimization problem for the best allocating strategy. From these counterfactual exercises we further contribute to the understanding of which police patrol deployment strategies are more efficient. Our results are in line with [14], but since we estimate a structural model we are able to evaluate counterfactual scenarios that significantly reduce crime.

The rest of the paper is organized as follows. In the next subsection we summarize the relevant Related Literature and frame our work within the literature investigating the causal effect of police presence on crime events. The Materials and Methods section details the spatial discrete-choice model we use to model criminals' choices of where to commit crimes. Thereafter, we present our Empirical Strategy to estimate the parameters of interest along with our IV approach to identify the causal effect of police exposure on crime, while the Data section

describes the experimental dataset used. In the Results section we analyze the estimations obtained regarding the causal effect of police presence on the level of crime, and the own- and cross- elasticities of such effect for each region of the city. In this section we use the estimated structural model to compare different counterfactual patrolling strategies. The Discussion section summarizes our findings and discusses limitations and next steps of our work.

## Related literature

Our work is mainly related to the literature on the causal effects of proactive police exposure on criminal activity. The evidence on such causal effect is mixed (see [13] for a comprehensive study for the US). [15] used meta-analytic techniques to assess the impact of disorder policing on crime. They identified 30 randomized experimental and quasi-experimental tests of disorder policing suggesting that policing disorder strategies are associated with an overall statistically significant modest crime reduction. [16] conducted a systematic review and a meta-analysis examining the extent to which there is crime displacement or a diffusion of crime in medium-sized or large geographic areas. They reviewed 19 publications covering 20 quasi-experimental studies. They found no significant overall evidence of displacement or a diffusion of benefits. [17] reported the results of a randomized controlled trial of police effectiveness across 60 violent crime hot spots in Philadelphia. Their results suggested a significant reduction in the level of violent crime for the treated area after 12 weeks. Moreover, they showed that targeted areas in the top 40% of pre-treatment violent crime counts (i.e., the most critical regions) had significantly less violent crime during the operational period. Targeted areas outperformed the control sites by 23%, resulting in a total net effect (once displacement was considered) of 53 violent crimes prevented. [18] examined the effectiveness of foot patrol in violent micro-places in Kansas City. They studied the effects of deployed foot patrol in hot spots for a period of 90 days. They employed a quasi-experimental design comparing four treatment to four control areas, estimating panel-specific autoregressive models for 30 weeks prior to and 40 weeks after the treatment. Their results reveal statistically significant short-run reductions in violent crime in the micro-places receiving foot patrol treatment, while no such reductions were observed in the control areas. At the same time they found no evidence of crime displacement to spatially contiguous areas. Similarly, [19] study the spatial effect of police foot patrol in British Columbia, finding that the policy was effective in reducing crime when compared to the crime patterns in the same area before the program. This effect was concentrated in property crimes with no evidence of spatial crime displacement. In a recent study, [20] conducted a controlled field experiment of police placed-based interventions on violent crime. Their study spans a 12-month period of intervention in 0.5% of the city of Pittsburgh's area. They found statistically significant reductions in serious violent crime counts within treatment hot spots as compared to control hot spots, with an overall reduction of 25.3% in violent crimes such as homicides, rape, robbery, and aggravated assault. Only foot patrols, not car patrols, had statistically significant crime reductions in hot spots. They found no evidence of crime displacement, but a weakly statistically significant spillover of crime prevention benefits to adjacent areas.

In sharp contrast, some studies in Latin America found no or even negative effects of increased policing on crime levels. Among these, we highlight [14], a placed-based police and city services intervention at scale for Bogotá D.C., Colombia. The authors randomly assigned 1,919 streets to an 8-month treatment of doubled police patrols, greater municipal services, both, or neither. They studied the direct and spillover effects of such targeted state services. They found that increasing state presence has modest direct impacts, even when focusing on the highest-crime hot spots and with crime displaced nearby. Confidence intervals suggest

they can rule out total reductions in crime of more than 2%. In the same line, [21] study the effect of an increase in police patrol by randomly assigning 384 out of 967 hotspots in Medellín, Colombia, to a six-months experiment finding no direct effects of increased police presence on crime reduction (besides car thefts). Similarly, [22] study the effect on the level of crime of increased police presence in hotspots flagged by a predictive policing software in Montevideo, Uruguay. Using data from an RCT, the authors found no statistically significant reduction in the overall crime or robberies. Despite the widespread belief that police presence significantly reduces crime, and the positive evidence in the literature for large cities, this does not appear to be the case for major cities in Latin America.

Finally, we contribute in a lesser extent to the literature on discrete choice modeling of crime phenomena. The pioneer applications of spatial discrete choice modeling of crime are [23–25]. In [23], the authors studied the selection of crime (burglary) locations in the city of The Hague, Netherlands. They evaluate several interesting hypotheses about crime: Are more valuable properties more attractive to burglars? Do higher mobility, neighborhood ethnicity, distance to burglar's home, distance to city center, etc., have a causal effect on crime? They used a conditional logit model with individual sociodemographic data of 290 burglars who committed 548 burglaries in the city during the period 1996–2001. Their results support and quantify some of the working hypothesis but there is no causal identification of the effect of police patrolling on crime. Given that the the i.i.d. hypothesis used in [23] for maximum likelihood estimation is controversial due to spatial correlation, [25] studied the role of spatial resolution and alternative modeling strategies. Finally, [24] used a spatial choice model coupled with clustering techniques to estimate a mixture model of crime. Their model focuses on crime prediction and does not address the causal problem that we address in this paper.

## Materials and methods

### Proposed model

The model we estimate in this paper is an adaptation of the discrete-choice model of demand introduced by [26]. In his paper, [26] describes a –rather parsimonious– methodology that (1) allows one to estimate the structural parameters that drive the demand for differentiated products using aggregate data of product market shares, and (2) allows one to correct for the endogeneity of prices (and other product characteristics) by using Instrumental Variables (IV) and estimating by Two Stages Least Squares (TSLS). Thus, the discrete choice model we present in this paper is estimated using aggregate data of reported (and mainly unsolved) crimes in different locations of Bogotá D.C., police presence in each of the locations, and some other characteristics of the location that might potentially affect crime presence in each of the locations. Our model is described next.

Consider *N* potential criminal offenders with *symmetric preferences* (this assumption can be relaxed, but it is not under the scope of this paper), each of them living in Bogotá D.C. (or in its surroundings) and deciding between *J*+ 1 locations in the city where to commit a crime. Each potential offender bases her location choice on her perceived utility of committing a crime in each of the *J*+ 1 locations. The associated utility $u_{ij}$ of agent *i* of selecting location *j* to commit a crime is given by

$$u_{ij} = \alpha P_j + X_j \beta + \xi_j + \varepsilon_{ij} \qquad (1)$$

where $P_j$ is a measure of the police presence in location *j*, $X_j$ is a vector of *K* observed characteristics of the location, $\xi_j$ is the constant term and might be thought as the mean utility associated to unobserved (by the researcher) characteristics of location *j*, $\alpha$ and $\beta$ are unknown

parameters that determine the influence of $P_j$ and $X_j$ on $u_{ij}$, respectively, and $\varepsilon_{ij}$ is the idiosyncratic error term.

It follows that the probability that potential offender $i$ selects location $j$ is given by

$$
\begin{aligned}
P(i \text{ chooses } j) \quad &= P(u_{ij} \geq u_{ik} \quad \forall \quad k \neq j) \\
&= P(u_{ij} \geq u_{i0}, \cdots, u_{ij} \geq u_{iJ}) \\
&= P(\delta_j + \varepsilon_{ij} \geq \delta_0 + \varepsilon_{i0}, \cdots, \\
&\qquad \delta_j + \varepsilon_{ij} \geq \delta_J + \varepsilon_{iJ}) \\
&= P(\varepsilon_{ij} - \varepsilon_{i0} \geq -(\delta_j - \delta_0), \cdots, \\
&\qquad \varepsilon_{ij} - \varepsilon_{iJ} \geq -(\delta_j - \delta_J))
\end{aligned}
\tag{2}
$$

where $\delta_j = \alpha P_j + X_j \beta + \xi_j$ and $j = 0$ corresponds to the *outside option* of not committing a crime in any of the $J$ locations of the city. We denote the probability described by Eq (2) as $s_{ij}(P_j, X_j, \xi_j; \alpha, \beta)$.

Assuming that all the $\varepsilon_{ij}$ terms are i.i.d and follow an extreme value type I distribution, we have, for any pair of different locations $j$ and $j'$, that $\varepsilon_{ij} - \varepsilon_{ij'}$ follows a logistic distribution and, by Eq (1), location choice probabilities have a closed-form expression given by:

$$
s_{ij}(P_j, X_j, \xi_j; \alpha, \beta) = \frac{\exp(\delta_j)}{1 + \sum_{k=1}^{J} \exp(\delta_k)}
\tag{3}
$$

where, again, option $j = 0$ is assumed to be the *outside option* and, thus, $\delta_0$ is normalized to zero. The final logistic specification may be questionable since it implies the well-known assumption of independence of irrelevant alternatives (IIA): the relative odds of choosing location $i$ over $j$ is the same no matter what other alternatives are available. To test for the validity of this assumption, we follow [27]'s specification test; the results are reported in the Figure in S1 Fig. This *outside option* in this particular context can be interpreted as the choice of not committing a crime at all, or committing a crime but not at any of the street segments of our data set. Therefore, it can include some other choices such as committing a crime different from those we have in our data set such as a cybernetic crime or committing a crime at another (nearby) city or town.

From Eq (3) it follows that the share of committed crimes at location $j$ is:

$$
\begin{aligned}
S_j(P_j, X_j, \xi_j; \alpha, \beta) \quad &= \int_i s_{ij}(P_j, X_j, \xi_j; \alpha, \beta)\varphi(i)di \\
&= s_{ij}(P_j, X_j, \xi_j; \alpha, \beta) \int_i \varphi(i)di \\
&= s_{ij}(P_j, X_j, \xi_j; \alpha, \beta) \\
&= \frac{\exp(\delta_j)}{1 + \sum_{k=1}^{J} \exp(\delta_k)}.
\end{aligned}
\tag{4}
$$

This equivalence between $S_j(P_j, X_j, \xi_j; \alpha, \beta)$ and $s_{ij}(P_j, X_j, \xi_j; \alpha, \beta)$ is a result of the symmetric preferences assumption: $s_{ij}(P_j, X_j, \xi_j; \alpha, \beta)$ does not depend on individual characteristics.

An advantage of this model is that, contrary to the traditionally implemented Poisson Model in target-based studies of crime location, it allows us to estimate own- and cross-elasticities in a fairly direct way (see [28] for a detailed explanation.) Particularly, from Eq (4) we can easily derive the own- and cross-elasticities of crime with respect to police presence $P_j$ (or any observed characteristic $x_{rj} \in X_j$), which captures the percentage change in crime when police

presence is increased by 1%. In particular, the derivatives with respect to $P_j$ are given by:

$$\frac{\partial S_j}{\partial P_\ell} = \begin{cases} \alpha S_j(1 - S_j) & \text{if } j = \ell \\ -\alpha S_j S_\ell & \text{if } j \neq \ell \end{cases} \tag{5}$$

and thus, the police own- and cross-elasticities of crime are:

$$E_{S_j, P_\ell} \equiv \frac{\partial S_j}{\partial P_\ell} \frac{P_\ell}{S_j} = \begin{cases} \alpha(1 - S_j)P_j & \text{if } j = \ell \\ -\alpha S_\ell P_\ell & \text{if } j \neq \ell \end{cases}. \tag{6}$$

The model described in this section contrasts both with the Conditional Logit (CL) model, which is commonly used in the crime location discrete choice framework, and the Poisson Model, mainly used in target-based crime location studies [29, 30]. Particularly, this model can be thought of as an aggregated version of the traditional Conditional Logit model that holds the advantages of the former and the latter. Regarding the advantages of the CL model, our model is also based on Random Utility Maximization (RUM) theory. This allows us to directly derive the statistical model to be estimated and, therefore, recover the structural parameters that drive the crime location choice [26, 31].

On the other hand, our model does not represent a "zero-sum" world where one more crime in one location represents exactly one less crime in another location, as is the case of the CL model. This means that our model allows for changes in attributes (such as police presence) to result in an overall change in the city's level of crime, which is an advantage of the Poisson Model. Additionally, in line with the traditional CL model and in contrast with the Poisson model, our model allows for own-elasticities not to be constant and cross-elasticities to be different from zero (see [28] for a detailed discussion). Lastly, our model can be easily estimated by Two Stages Least Squares (TSLS), which, in turn, allows us to deal with endogeneity with relative ease. That is not the case with Maximum Likelihood, which is more "parametrically" restrictive.

## Empirical strategy

**Main specification.** To estimate the structural parameters $\theta = (\alpha, \beta)$ from Eq (1) we follow [26] and note that:

$$\delta_j \equiv \log(S_j) - \log(S_0) \quad = \alpha P_j + X_j \beta + \xi_j, \tag{7}$$

and thus Eq (7) represents our ideal specification. In this equation, the definitions of the variables are the same as those mentioned in the Proposed Model Section. Namely, $P_j$ is a measure of police presence at location $j$ and $X_j$ is the set of measured characteristics for the same location (including an all-ones vector). $\xi_j$, in this case, represents the error term.

On the other hand, $S_j$ for $j \in \{1, \cdots, J\}$ is defined as the proportion of crimes committed at location $j$. That is,

$$S_j = \frac{C_j}{N} \tag{8}$$

where $C_j$ is the number of crimes committed at location $j$ and $N$ is the number of potential offenders. $S_0$, on the contrary, is not directly observed since it captures the choice of the *outside*

*option*—i.e., not committing a crime in any of the locations. However, we exploit the fact that

$$\sum_{j=0}^{J} S_j = 1. \tag{9}$$

And, therefore, we recover $S_0$ as

$$S_0 = 1 - \sum_{j=1}^{J} S_j \equiv 1 - \sum_{j=1}^{J} \frac{C_j}{N}. \tag{10}$$

Now that we know all the components from Eq (7), we can easily estimate it by OLS. However, the OLS estimation faces one problem: $P_j$ is endogenous. In particular, under the assumption that the police force is assigned according to the number of committed crimes in each of the locations, there exists simultaneity between $\delta_j$ and $P_j$. $\delta_j$ is not directly a measure of crimes but it is a function of them. So, if there exists simultaneity between $C_j$ and $P_j$, there exists simultaneity between $\delta_j$ and $P_j$. Thus, OLS estimates from Eq (7) are plausibly biased and inconsistent.

**Dealing with endogeneity: Two-Stage Least Squares (TSLS).** In 2016, [14], along with the National Police of Colombia and the Mayor's Office of Bogotá, designed and implemented a multi-arm security experiment at the level of street segments. Two different types of intervention were randomly delivered: 1) an increased police patrol time and 2) an improvement of the delivery of city services (street lighting and cleanup). Starting in January 2016 and during 8 months, 756 out of 1,919 street segments labeled as crime hot spots –out of the 136,984 street segments of the city– received a doubled patrolling time (92–167 minutes of police patrol per day) [14]. Also, in March 2016, 201 of the 1,919 hot spots received more intensive street light repair and cleaning [14] (for further information about the different treatment arms see [14]). In this paper, we exploit the random assignment of the first type of treatment to instrument the police presence $P_j$, identify the structural parameters of interest, and consistently estimate them by TSLS. We take $X_j$ as given and exogenously determined. Thus, we do not instrument them.

Following [32, 33], the needed assumptions for the treatment assignment, $Z$, to be a valid instrument for $P_j$ are: 1) independence, 2) exclusion restriction, 3) rank condition (relevance), and 4) monotonicity (no defiers). First, given that the police patrol treatment was randomly assigned within the hot spot street segments, $Z$ is plausibly independent of the potential outcomes (crime shares ratios) and the potential treatment status (police patrol time) once it is controlled for the hot spot label status (i.e., $1\{j \text{ is a crime hot spot}\} \in X_j$) [14]. Second, since the treatment arm that we consider in this paper only determines police patrol time, it is plausible that $Z$ only affects $\delta_j$ through $P_j$. Third, according to the findings of [14] the *"police complied with their new orders for the full 8 months."* Thus, the rank condition is plausible and has been already verified. Fourth, given that the police officers were monitored via GPS every 30 seconds and police officers (and workers in general) plausibly double their efforts in a task only when they are ordered or incentivized to do so, it is reasonable that the monotonicity condition holds. That is, only treated segments received increased patrol time. Control segments remained the same. Furthermore, the experimental design in [14] intended that the patrol time of untreated streets remained the same, which was empirically verified by the authors. Thus, under the [14] setting, $Z$ is a valid instrument for $P_j$, when each $j$ corresponds to a different street segment of the city.

As is explained in the Data Section, however, we do not estimate our model with street-level data. Instead, we use quadrant-level data, which represents a higher level of aggregation

unit. In this sense, the instrument we exploit for $P_j$ is the proportion of street segments within quadrant $j$ that were treated. Controlling now for the proportion of hot spots street segments within quadrant $j$, the assumptions that we discussed in the previous paragraphs naturally extend to this aggregated case and plausibly hold. Since the impact of the instrument on $P_j$ depends on the initial level of police presence, we additionally control for the baseline (i.e., before the experiment) level of police presence to avoid possible endogenous relationships.

Under the aforementioned assumptions, which, as was discussed, seem to hold, the IV methodology allows us to correctly identify the parameter vector $\theta$ and to consistently estimate it by TSLS. In this case, the parameter $\alpha$ –which is our main focus in this paper– captures a weighted average of all the possible per-unit causal responses of $\delta_j$ to a marginal change in $P_j$ caused by $Z$ ([33]). For more technical details check [33].

**Selecting location characteristics—Double selection.** In general, city locations might have several characteristics that can influence the presence of crime. When the set of available variables that might potentially influence crime location is vast, including all of them in the model to be estimated might result in power loss issues and high multicollinearity. This, in turn, might reduce the traceability of what really matters for the crime location choice. A natural solution to this problem is to manually select the covariates to be included. However, this might be cumbersome when the list of available variables is large, and also might lack rigor since is a subjective decision. To avoid this issue, we follow [34] and implement the double selection methodology to select the variables that should be included in $X_j$.

Following [34], we first (separately) run a regularized lasso over the following two equations

$$\delta_j = \tilde{X}_j \gamma + \mu_j, \tag{11}$$

$$P_j = \tilde{X}_j \vartheta + \lambda_j, \tag{12}$$

where $\delta_j$ and $P_j$ are the dependent and independent variables from Eq (7), respectively, and $\tilde{X}_j$ is a vector of variables that includes all the available and exogenous location features that potentially affect crime location. $\gamma$ and $\vartheta$ are vectors of coefficients associated with $\tilde{X}_j$ in each equation, while $\mu_j$ and $\lambda_j$ are the error terms.

Lasso algorithm applied to Eqs (11) and (12) selects the relevant location features that better predict the outcome variable $\delta_j$ and the police presence $P_j$, respectively. Given the selected variables from both estimations, we then estimate Eq (7) by TSLS using $X_j = \tilde{X}_j^{DS,\delta} \cup \tilde{X}_j^{DS,P}$, where $\tilde{X}_j^{DS,\delta}$ and $\tilde{X}_j^{DS,P}$ are the set of Lasso-selected location characteristics from Eqs (11) and (12).

## Data

In this work, we use the data set used by [14] in their study. In particular, we use their data on violent, property, and total criminal official records at the quadrant level (i.e., an administrative and irregular subdivision of the city used for assigning police shifts; two police officers per quadrant per shift) that occurred in Bogotá D.C. during the experiment ran by them in 2016. We also use information of the following characteristics at the quadrant level: proportion of paved street segments, proportion of street segments in use by industry and commerce, proportion of street segments in use by housing, proportion of low income street segments, proportion of middle income street segments, proportion of high income street segments, average distance from each street segment to the nearest shopping center, average distance from each street segment to the nearest education center, average distance from each street segment to the nearest park or recreational center, average distance from each street segment to the nearest religious center, average distance from each street segment to the nearest health center,

average distance from each street segment to the nearest services center (e.g., justice), average length of street segments, the average built meters per meter of street segment of length 100 meters around each street segment, and the proportion of street segments labeled as crime hot spots. Lastly, we take information on the average minutes of police patrol time that each segment received during the intervention and the proportion of street segments assigned to increased patrol time within each quadrant.

Provided this data set, we estimate three versions of Eq (7): one for violent crimes, another for property crimes, and the last one for total crimes. To construct each crime-specific $\delta_j$, we assume that the number of potential criminal offenders is given by the estimated number of unemployed (actively job-seeking or inactive) people aged 12 to 60 in the period of interest. $N$ is calculated using the public data of unemployment and population projections of the (Colombian) National Administrative Statistics Department (DANE from its Spanish initials). $N$ is $\approx$791, 223. The Data for this study is available at https://dane.gov.co/index.php/estadisticas-por-tema/mercado-laboral/empleo-y-desempleo/mercado-laboral-historicos#marco-2005 and https://www.dane.gov.co/index.php/estadisticas-por-tema/demografia-y-poblacion/proyecciones-de-poblacion. On the other hand, $P_j$ is given by average minutes of police patrol time (from [14]), $Z_j$ is the proportion of treated street segments, and $X_j$ is selected from the list of variables mentioned above.

Descriptive statistics of these variables at the quadrant level (1,050 spatial units) are presented in Table 1. Panel A reports descriptive statistics of the reported crime within each quadrant during 2016, panel B reports descriptive statistics of the average police patrol time in minutes within each quadrant during and before the intervention, and panel C reports descriptive statistics of the average characteristics of the street segments within each quadrant. As can be seen, property crimes seem to be more frequent than violent crimes. On average there were reported 11.84 violent crimes, 23.86 property crimes, and a total of 35.07 crimes. These types of crimes ranged between 0 to 70, 0 to 149, and 0 to 157, respectively. This depicts the high heterogeneity that exists in terms of security in Bogotá D.C. Panel B, on the other hand, shows that on average the patrol time received by the street segments within each quadrant during the intervention is about 46 minutes. However, this time varies from 4.98 to 711.23. This again reflects the high heterogeneity of the security in the city. Also, this contrasts with the average patrol time at baseline, which is about 19.68 minutes (42% of the time during intervention). It seems that the intervention did effectively increase police presence. Lastly, panel C reports summary statistics of the variables to be added to the vector $X_j$.

Fig 1 reports scatter plots –and their respective linear regression fit line– of the logarithm of the reported crimes against the logarithm of the police patrol time. This can be thought of as a first *naive* approximation to the estimation of the police elasticity of crime. The figure shows no relationship between police presence and any of the reported crimes. OLS estimates are -0.02 for violent crimes, 0.05 for property crimes, and 0.01 for total crimes. None of them is statistically significant. This counterintuitive result suggests that, if any, the police elasticity of crime is positive. These results, however, might be driven by the simultaneity that exists between crimes and police presence, and therefore, justify the need to account for endogeneity as was discussed in the Empirical Strategy section.

All code used in this study is available at: https://github.com/lgomezt/A-Location-Discrete-Choice-Model-of-Crime/tree/main

## Results

We divide our results in two main subsections. In the first subsection, we present the results of the estimation of our discrete choice model of crime (i.e., structural model). In the second

**Table 1. Descriptive statistics of Bogotá D.C.'s quadrants (2016).**

| | N | Mean | SD | Min | Max |
|---|---|---|---|---|---|
| *A. Reported Crimes* | | | | | |
| Violent Crimes | 1,050 | 11.84 | 8.56 | 0.00 | 70.00 |
| Property Crimes | 1,050 | 23.86 | 17.88 | 0.00 | 149.00 |
| Total Crimes | 1,050 | 35.70 | 21.42 | 0.00 | 157.00 |
| *B. Police Presence* | | | | | |
| Avg. police patrol time (minutes) | 1,050 | 46.58 | 52.09 | 4.98 | 711.23 |
| Baseline avg. police patrol time (minutes) | 1,050 | 19.68 | 18.47 | 3.79 | 248.03 |
| *C. Quadrant characteristics* | | | | | |
| Avg. longitude of street segments (mt) | 1,050 | 64.20 | 23.44 | 21.52 | 263.38 |
| Avg. built squared meters per street segment meter | 1,050 | 22,786.17 | 13,602.68 | 3.19 | 148,750.19 |
| Avg. distance to nearest shopping center | 1,050 | 739.00 | 706.64 | 21.72 | 4,686.01 |
| Avg. distance to nearest educational center | 1,050 | 319.18 | 234.42 | 55.98 | 2,767.30 |
| Avg. distance to nearest park/recreational center | 1,050 | 680.85 | 390.99 | 37.57 | 2,855.38 |
| Avg. distance to nearest police station | 1,050 | 603.29 | 362.46 | 60.47 | 2,943.86 |
| Avg. distance to nearest religious/cultural center | 1,050 | 501.72 | 365.22 | 28.90 | 3,829.53 |
| Avg. distance to nearest health center | 1,050 | 951.65 | 715.99 | 55.56 | 7,499.75 |
| Avg. distance to nearest additional services office | 1,050 | 698.72 | 575.83 | 39.38 | 5,193.83 |
| Avg. distance to nearest transport infrastructure | 1,050 | 102.69 | 65.70 | 17.13 | 1,091.14 |
| Avg. distance to closest surveillance camera | 1,050 | 592.81 | 522.62 | 17.72 | 6,051.21 |
| Prop. of street segments zoned for industry/commerce | 1,050 | 0.25 | 0.22 | 0.00 | 1.00 |
| Prop. of street segments zoned for housing | 1,050 | 0.66 | 0.27 | 0.00 | 1.00 |
| Prop. of middle income street segments | 1,050 | 0.45 | 0.48 | 0.00 | 1.00 |
| Prop. of high income street segments | 1,050 | 0.08 | 0.26 | 0.00 | 1.00 |
| Prop. of hot spot stret segments | 1,050 | 0.03 | 0.06 | 0.00 | 1.00 |
| Prop. of treated hot spot street segments | 1,050 | 0.01 | 0.03 | 0.00 | 0.50 |

Notes: Variables at the quadrant level were constructed from the aggregation or averaging of the original data set variables at the level of street segment. Reported crimes were aggregated. The remaining variables were averaged. Thus, summary statistics for these latter types of variables can be thought as *the mean/SD/min/max of the average* of the variable. An average of a dichotomous variable is reported as a proportion. Distances measured in meters. Avg.: Average. Prop.: Proportion. SD: Standard Deviation. Min: Minimum. Max: Maximum.

subsection, we use our estimated structural model of crime to study the crime consequences of some counterfactual scenarios of police allocation.

## Structural model

**$\alpha$ estimates.** Table 2 presents, for each of the three types of crimes, the TSLS estimates of $\alpha$ from four different specifications of Eq (7). The first specification corresponds to one where no covariates were included in $X_j$ apart from the proportion of hot spots in the quadrant, which is needed for the instrument to be valid. In the second specification, we add the quadrant characteristics that were selected by the Double Selection algorithm [34]. In the third one, we additionally control for the baseline police patrol time, and in the fourth one, we add locality fixed effect. A locality is the greater administrative division of Bogotá D.C., and each of them contains several quadrants within. It is known that the patterns of crime in each locality are different, and thus, by including the locality fixed effects, we control for those differential patterns.

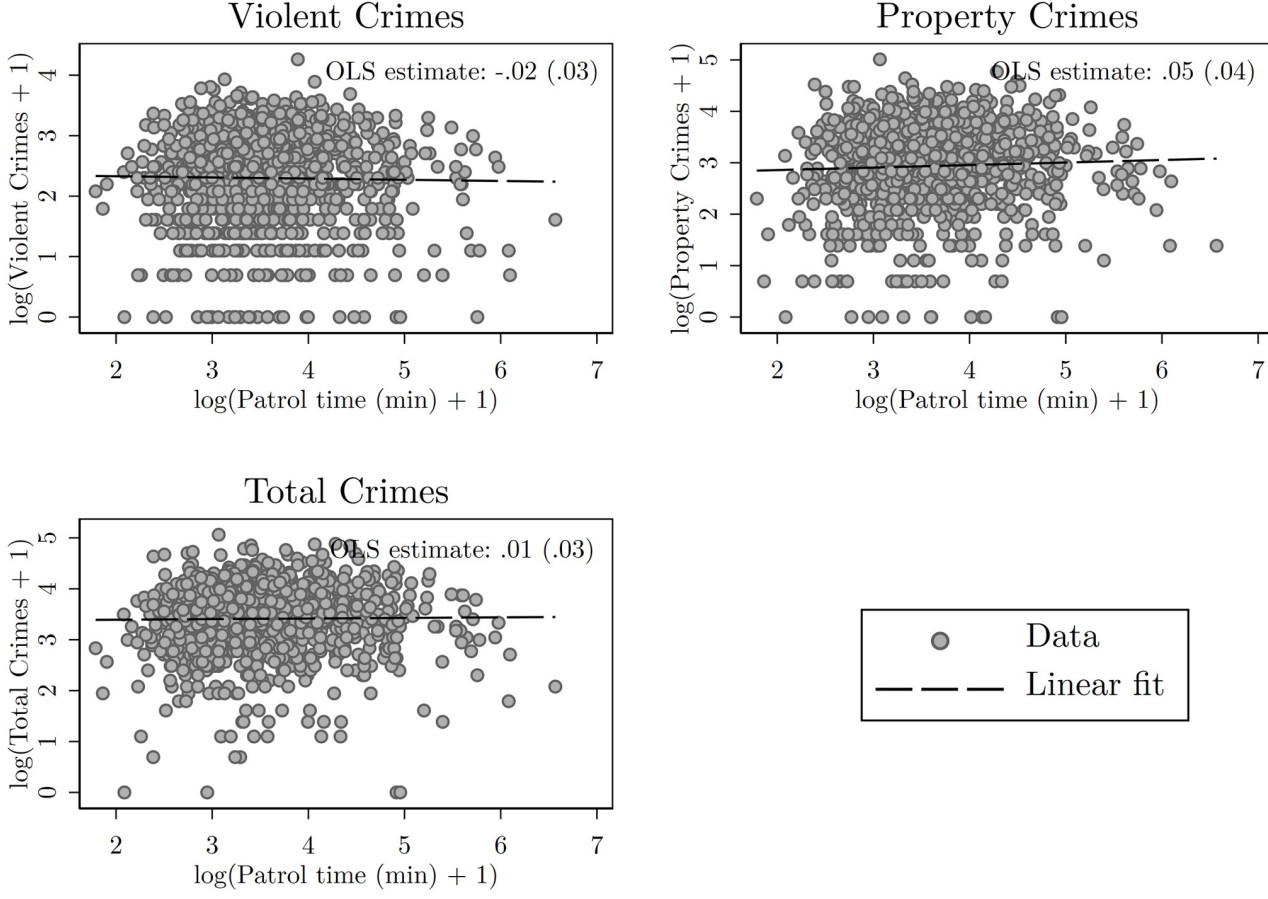

**Fig 1. Scatter plots of the logarithm of each type of crime against the logarithm of the average patrol time within each quadrant.** Linear fit displayed by the black dashed line. These graphs display a naive calculation of the police elasticity of crime. OLS coefficient estimates and their standard errors in parentheses reported at the top-right corner of each graph. *p<0.1; **p<0.05; ***p<0.01. Source: Own elaboration.

**Table 2. TSLS $\alpha$ estimates for the discrete spatial location choice model.**

|  | Violent crimes | | | | Property crimes | | | | Total crimes | | | |
|---|---|---|---|---|---|---|---|---|---|---|---|---|
|  | (1) | (2) | (3) | (4) | (5) | (6) | (7) | (8) | (9) | (10) | (11) | (12) |
| $\alpha$ | -0.004* | -0.003* | -0.005* | -0.006** | -0.004** | -0.005*** | -0.009** | -0.008* | -0.005** | -0.005*** | -0.008** | -0.008* |
|  | (0.002) | (0.001) | (0.003) | (0.003) | (0.002) | (0.002) | (0.004) | (0.005) | (0.002) | (0.002) | (0.004) | (0.004) |
| Observations | 1,050 | 1,050 | 1,050 | 1,050 | 1,050 | 1,050 | 1,050 | 1,050 | 1,050 | 1,050 | 1,050 | 1,050 |
| Controls | No | Yes | Yes | Yes | No | Yes | Yes | Yes | No | Yes | Yes | Yes |
| Past police presence | No | No | Yes | Yes | No | No | Yes | Yes | No | No | Yes | Yes |
| Locality FE | No | No | No | Yes | No | No | No | Yes | No | No | No | Yes |
| R-squared | -0.023 | 0.309 | 0.310 | 0.380 | 0.000 | 0.339 | 0.340 | 0.414 | -0.035 | 0.187 | 0.190 | 0.281 |

Notes:

*** p<0.01,

** p<0.05,

* p<0.1. Cluster robust standard errors at the level of locality in parentheses.

As can be seen, the estimates are fairly consistent across specifications, and thus, we interpret the fourth one—our preferred specification. We find that police presence in a certain location has, on average, a negative impact on the utility of committing either a violent, a property, or any crime in it. An increase in one minute of police patrol time in location $j$ reduces, on average, 0.006 units the utility of committing a violent crime and 0.008 units the utility of committing either property or any crime in that location. These effects are statistically significant at the 5% for violent crimes and 10% for property and total crimes. Thus, it follows from these estimates that police presence reduces the probability of a potential criminal committing a crime in a protected location.

According to [35], the estimates of a traditional Conditional Logit model are equivalent to those of a Poisson Model with aggregated data. This is true since the first-order conditions of the likelihood maximization of both models are equivalent [35]. Therefore, in order to compare our model to the Poisson and the traditional CL logit model, we additionally estimate a target-based Poisson Model using the number of reported crimes in each quadrant as the dependent variable. We, again, deal with the endogeneity by instrumenting $P_j$ with the treatment assignment condition. However, in this case, estimates are recovered either by the Generalized Method of Moments (GMM) or the Control Functions (CF) Methodology (see [36], chapters 8, 9, and 21 for technical details). Table 3 presents the $\alpha$ estimates of this Poisson Model. As can be seen, the estimates are very similar to those presented in Table 2. This similarity aligns with the fact that our model is an aggregated version of the CL model, and suggests that the methodology we implement in this paper is correct.

Additionally, it is to be noted that when TSLS estimates are invalid, OLS estimates provide suggestive information about the true impact of police presence on crime. In particular, note that, given the simultaneity that exists between crime and police presence, OLS estimates are downward biased. Note that if the impact of police on crime is negative and the impact of crime on police is positive, OLS estimates that ignore the simultaneity just combine both effects into one. That is, a positive and a negative effect are averaged, yielding a less negative (or even a positive) result. Therefore, if we estimate Eq (7) by OLS we would obtain lower bounds (in absolute value) of the real average impact of police presence on the—and thus on the occurrence– of committing a crime in a certain location. As can be seen in the Table in S1 Table, this holds and, therefore, provides confidence to our principal estimates.

**Table 3. $\alpha$ estimates from a target-based poisson regression model.**

| | Violent crimes | | | | Property crimes | | | | Total crimes | | | |
|---|---|---|---|---|---|---|---|---|---|---|---|---|
| | (1) | (2) | (3) | (4) | (5) | (6) | (7) | (8) | (9) | (10) | (11) | (12) |
| $\alpha$ | -0.004 | -0.003** | -0.005* | -0.006** | -0.007 | -0.006*** | -0.010** | -0.010* | -0.006* | -0.005*** | -0.008*** | -0.008** |
| | (0.003) | (0.001) | (0.002) | (0.003) | (0.005) | (0.002) | (0.004) | (0.005) | (0.003) | (0.002) | (0.003) | (0.004) |
| Observations | 1,050 | 1,050 | 1,050 | 1,050 | 1,050 | 1,050 | 1,050 | 1,050 | 1,050 | 1,050 | 1,050 | 1,050 |
| Controls | Yes | Yes | Yes | Yes | Yes | Yes | Yes | Yes | Yes | Yes | Yes | Yes |
| Previos patrol time | No | No | Yes | Yes | No | No | Yes | Yes | No | No | Yes | Yes |
| Locality FE | No | No | No | Yes | No | No | No | Yes | No | No | No | Yes |
| Method | GMM | CF | CF | CF | GMM | CF | CF | CF | GMM | CF | CF | CF |

Notes:

*** p<0.01,

** p<0.05,

* p<0.1. Cluster robust standard errors at the level of locality in parentheses.

Lastly, note that in this work we assume that the number of potential criminals is given by the total number of unemployed people aged 12 to 60, which results in a conservative share of crimes in each of the city locations. Thus, we are confident that the reported TSLS estimates, which are relatively close to the OLS estimates, are informative enough about the causal impact of police presence (and the rest of covariates) on crime location choice and, therefore, general conclusions can be obtained from them.

**Selected covariates.** Table 4 presents the coefficients associated with the covariates that were selected and ultimately included in our model estimation. Surprisingly, the Double Selection algorithm selected all the available control variables. Following our model, the coefficients depicted in the table reflect which attributes make a location more or less attractive to commit a crime. In general, the results are quite intuitive. Concretely, the coefficients on the average distance to a shopping center, education center, services offices, and nearest transportation infrastructure are negative, and the coefficients associated with building presence and housing presence are positive but small. These might reflect the fact that close to these places there is usually more mobility and human interaction, and thus, greater opportunity or ease for criminals to commit a crime.

In contrast, the coefficient on the average distance to a religious center is positive suggesting the relevance of other determinants of crime associated with religious beliefs and respect for certain norms. As expected, criminals are less likely to select a high-income location to commit a violent crime, but are more likely to select a middle- or high-income to commit a property crime. We interpret these results as follows: the perceived benefit from a property crime is highest in richer places, while the perceived risk of committing a violent crime in such places is also higher. Therefore, violent crimes move towards poorer areas, while property crimes to richer ones.

Additionally, we find a negative coefficient for the average distance to the nearest police station, a positive coefficient for the proportion of hot spots within the quadrant, and a null coefficient for the average distance to the closest surveillance camera. The first result, although counterintuitive, might reflect the fact that the police stations are located in historically insecure places and that it is not the infrastructure itself, but the effective police presence that can affect crime levels. The second result reflects the fact that criminals tend to commit crimes in insecure places, and that there is constant feedback between past and future crime locations: crimes are more common where past crimes have been committed. The third result suggests that surveillance cameras have a low effect in deterring criminals to commit crimes in the watched places.

Lastly, no statistically significant effects are found for the average distance to parks or health centers, the average longitude of street segments, nor the proportion of street segments zoned for industry and commerce.

**The police elasticity of crime.** Our most important results from a public policy perspective are the estimation of the own- and cross-elasticities of crime to patrolling time at different locations, which captures the percentage change in crime that results from a 1% increase in the police patrol time that a location receives. To the extent of our knowledge, the estimation of these statistics in a properly identified structural model of crime location is new in the literature. Given the estimated parameters from our preferred specification, we follow Eq (6) to calculate own- and cross-elasticities of crime with respect to the police presence (police elasticities of crime, from now on).

The first panel of Fig 2 reports the own- elasticity distribution across all locations. For violent crimes, the elasticity is, on average, −0.26 and is statistically significant with 95% confidence. That is, a 10% increase in patrolling time in a quadrant reduces crime, on average across all quadrants, by 2.6% in that quadrant. The effect on property crime is almost 1.5 times as large, which seems consistent with intuition. This result, therefore, holds for total crimes.

The second panel of Fig 2 shows the results in terms of crime displacement. By construction, our model takes every crime location as a substitute for the others. This is clear in Eq (6), where the cross-elasticity has a positive sign and, by construction, a police presence increase in one location results in a crime displacement to other locations. Hence, our model is biased in

**Table 4. TSLS $\beta$ estimates of the selected variables.**

|  | Violent crimes | Property crimes | Total crimes |
|---|---|---|---|
|  | (1) | (2) | (3) |
| Average built squared meters per street segment meter | 0.0000*** | 0.0000* | 0.0000*** |
|  | (0.0000) | (0.0000) | (0.0000) |
| Average distance to nearest shopping center | -0.0000 | -0.0003*** | -0.0001** |
|  | (0.0000) | (0.0001) | (0.0001) |
| Average distance to nearest educational center | -0.0003*** | 0.0001 | -0.0001 |
|  | (0.0001) | (0.0001) | (0.0001) |
| Average distance to nearest park/recreational center | 0.0001 | -0.0000 | -0.0000 |
|  | (0.0001) | (0.0001) | (0.0001) |
| Average distance to nearest police station | -0.0003*** | -0.0002* | -0.0002*** |
|  | (0.0001) | (0.0001) | (0.0001) |
| Average distance to nearest religious/cultural center | 0.0001** | 0.0001* | 0.0002*** |
|  | (0.0001) | (0.0001) | (0.0001) |
| Average distance to nearest health center | 0.0000 | -0.0000 | -0.0000 |
|  | (0.0001) | (0.0001) | (0.0001) |
| Average distance to nearest transport infrastructure | -0.0012*** | -0.0013** | -0.0014*** |
|  | (0.0004) | (0.0005) | (0.0005) |
| Proportion of street segments zoned for industry/commerce | 0.3433 | 0.3073 | 0.2764 |
|  | (0.3190) | (0.4207) | (0.3905) |
| Proportion of street segments zoned for housing | 0.4341* | -0.1778 | -0.0574 |
|  | (0.2582) | (0.2954) | (0.2868) |
| Proportion of middle income street segments | 0.0007 | 0.5815*** | 0.3855*** |
|  | (0.0827) | (0.1103) | (0.1009) |
| Proportion of high income street segments | -0.6294*** | 0.4551** | 0.1491 |
|  | (0.0832) | (0.2190) | (0.1757) |
| Average longitude of street segments (mt) | -0.0021 | -0.0012 | -0.0018 |
|  | (0.0014) | (0.0036) | (0.0027) |
| Average distance to nearest additional services office | -0.0000 | -0.0001** | -0.0001 |
|  | (0.0001) | (0.0001) | (0.0001) |
| Average distance to closest surveillance camera | 0.0000 | -0.0001 | -0.0001 |
|  | (0.0001) | (0.0001) | (0.0001) |
| Proportion of hotspot street segments | 0.6522** | 1.5188*** | 1.3312*** |
|  | (0.3028) | (0.5798) | (0.4215) |
| Observations | 1,050 | 1,050 | 1,050 |
| R-squared | 0.3801 | 0.4138 | 0.2813 |
| Controls | Yes | Yes | Yes |
| Previos patrol time | Yes | Yes | Yes |
| Locality FE | Yes | Yes | Yes |

Notes:

*** p<0.01,

** p<0.05,

* p<0.1. Cluster robust standard errors at the level of locality in parentheses.

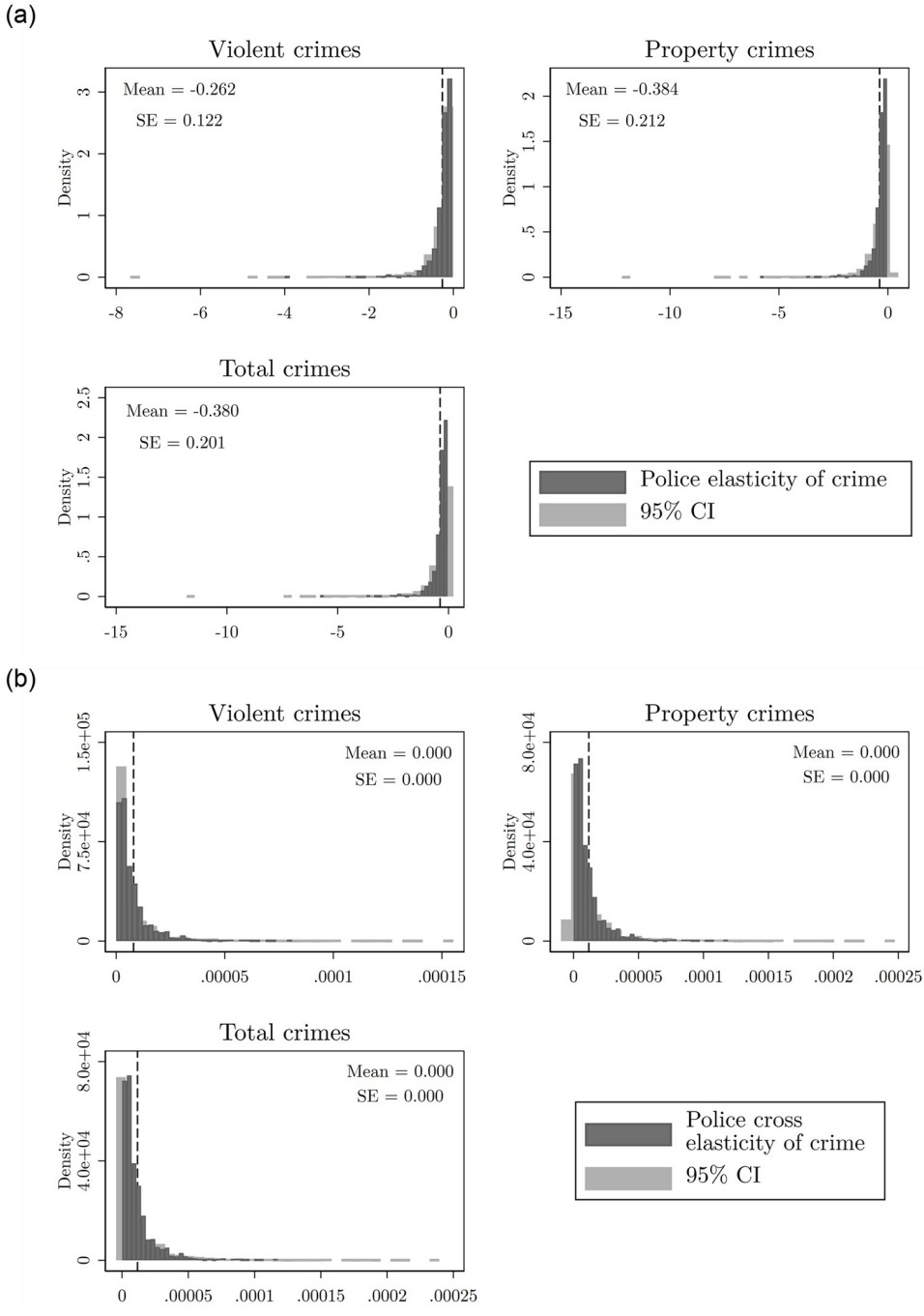

**Fig 2. The first panel of this figure reports the own- elasticity of crime distribution across all locations and all types of crime.** The second panel shows the cross-elasticities of crime distribution across all locations and all types of crime. Recall that the cross elasticity is a property of the location (the same cross elasticity with all other locations). (a) Police elasticity of crime, (b) Cross police elasticity of crime.

favor of a hypothesis that has been at the center of the discussion in the criminology literature: crime is displaced rather than reduced in response to police presence. However, by construction, crime displacement in our model is, controlling for police presence, equally likely to all crime locations. This is an oversimplifying assumption that implies cross-elasticities may be

very low although relevant. We discuss this further below when we compare our results with [14].

Our results are in line with those of found by [14]. In particular (see Table 3, page 2042 in [14]), they find that doubling the police patrol time causes an average reduction of crime of 13% on treated segments (direct effect). This is comparable to our police elasticity of crime (Panel (a), Fig 2). Now, they also report a small spillover effect (indirect effect) within street segments at less that 250 meters from treatment, suggesting that crime increases in the neighborhood of treated segments. This is comparable to our cross police elasticity (Panel (b), Fig 2). Although this indirect effect is small there are so many non-treated segments that are in the neighborhood of a treated segment, that when they aggregate and estimate the total effect on crime, the indirect effects override the direct effects and crime is estimated to increase (3%) as a consequence of police intervention. However, this aggregate result is not statistically significant at a 10% level. In the next section we show that our results are consistent with these results from [14]. However, since we estimate a structural model, we are able to perform other types of policy scenarios that we show have an important impact on crime reduction.

## Counterfactual policy scenarios

A great advantage of using structural models of social phenomena is the ability, conditional to the *prior* that the model is an accurate description of reality, to study alternative policy scenarios. The question we now want to answer is: What would have been the number of crimes had police patrols pursued different patrolling strategies? (1) uniform time (each segment receives 33.82 minutes of patrol time independently of any characteristic, it is the total observed patrol time divided by the number of quadrants); (2) time spent proportional to historic crime rates per segment; and (3) Police patrol time set to a minimum for all the quadrants and the residual time (i.e., the difference between the observed time and the minimum time) is reassigned such that quadrants that report both the highest levels of crime and the highest police elasticity of crime receive their initial police patrol time plus an $x$% increase, where $x \in \{0, 10, 20, 30, 40, 50, 60, 70, 80, 90, 100\}$. This additional time is indirectly reduced in quadrants that are both less insecure and less elastic, such that the total patrolling time across the city remains constant. For a detailed explanation of this reassignment strategy, see S1 Appendix. An important characteristic of all policy scenarios is that, by construction, the economic costs of all of them is roughly the same as the observed scenario (see Table 5: Observed scenario), the current police patrolling strategy. Since the outcome is measured in terms of their average effect on crime, then our policy scenarios measure efficiency gains, due to a better police-time reassignment, on the city. We leave for a separate subsection the estimation of the optimal reassignment policy and how it compares to [14] study.

**Basic policies.** Table 5 shows the average observed crimes (i.e., Observed row) and the in-sample fit (i.e., Predicted row) of the model in terms of the average predicted crimes, respectively. As can be seen, our model underestimates by approximately three events the average occurrence of crimes, and thus, can be regarded as a conservative and good enough representation of reality.

The rest of the rows form Table 5 show what would have happened had the police been deployed under the four different policy scenarios. In the first place, the uniform-policing time distribution, which is the uninformed decision, results in more crime occurrences (compared to the predicted number of actual crimes). In the second place, proportional time results in basically the same levels of crime as those predicted by the model in the base scenario. We argue, this is likely the strategy most frequently used by the police in the city of Bogotá to allocate patrolling time.

**Table 5. Counterfactual policy scenarios.**

| | Violent Crimes | | | Property Crimes | | | Total Crimes | | |
|---|---|---|---|---|---|---|---|---|---|
| | Benefited Q. | | Predicted # | Benefited Q. | | Predicted # | Benefited Q. | | Predicted # |
| | N | % | Mean (SD) [SE] | N | % | Mean (SD) [SE] | N | % | Mean (SD) [SE] |
| Observed | - | - | 11.84 | - | - | 23.86 | - | - | 35.70 |
| | | | (8.56) | | | (17.88) | | | (21.42) |
| Predicted | - | - | 11.14 | - | - | 21.58 | - | - | 32.59 |
| | | | [0.91] | | | [1.75] | | | [1.89] |
| Uniform time | - | - | 11.35 | - | - | 23.80 | - | - | 34.67 |
| | | | [1.61] | | | [2.54] | | | [2.54] |
| Proportional time | - | - | 11.83 | - | - | 22.19 | - | - | 33.92 |
| | | | [1.84] | | | [2.38] | | | [2.67] |
| 10% increase | 1,000 | 95.2 | 11.10 | 1,000 | 95.2 | 21.41 | 999 | 95.1 | 32.36 |
| | | | [0.90] | | | [1.71] | | | [1.81] |
| 20% increase | 914 | 87.0 | 10.15 | 911 | 86.8 | 21.80 | 912 | 86.9 | 31.22 |
| | | | [0.97] | | | [1.66] | | | [1.73] |
| 30% increase | 855 | 81.4 | 10.30 | 854 | 81.3 | 22.42 | 854 | 81.3 | 31.53 |
| | | | [1.07] | | | [1.64] | | | [1.68] |
| 40% increase | 803 | 76.5 | 10.40 | 805 | 76.7 | 22.50 | 805 | 76.7 | 31.36 |
| | | | [1.13] | | | [1.66] | | | [1.67] |
| 50% increase | 756 | 72.0 | 10.46 | 756 | 72.0 | 22.77 | 756 | 72.0 | 31.57 |
| | | | [1.28] | | | [1.80] | | | [1.82] |
| 60% increase | 717 | 68.3 | 10.54 | 717 | 68.3 | 23.14 | 715 | 68.1 | 32.03 |
| | | | [1.40] | | | [1.85] | | | [1.85] |
| 70% increase | 682 | 65.0 | 10.59 | 682 | 65.0 | 23.33 | 680 | 64.8 | 32.26 |
| | | | [1.50] | | | [1.87] | | | [1.89] |
| 80% increase | 645 | 61.4 | 10.67 | 641 | 61.0 | 23.65 | 643 | 61.2 | 32.51 |
| | | | [1.61] | | | [1.94] | | | [1.89] |
| 90% increase | 600 | 57.1 | 10.75 | 601 | 57.2 | 24.16 | 601 | 57.2 | 33.34 |
| | | | [1.73] | | | [2.11] | | | [2.07] |
| 100% increase | 568 | 54.1 | 10.82 | 568 | 54.1 | 25.21 | 569 | 54.2 | 34.18 |
| | | | [1.85] | | | [2.40] | | | [2.44] |

Notes: Mean observed and predicted crimes for each of the policy or time reassignment strategies of police patrol time. Standard errors at the level of locality in squared parentheses.

The third policy scenario leaves a couple of lessons. First, provided that there are limited time resources (i.e., the total available time used by all police officers patrolling the city), there exists a trade-off between the number of quadrants that can benefit from an increase in police patrolling time and the magnitude of the increase itself. That is, the greater the percentage increase in time, the fewer quadrants receive such increase. Second, given this trade-off, it is better to increase by small percentages the police patrol time of most of the quadrants than increasing by a great percentage the police patrol time of just a few quadrants. Specifically, it seems that an effective way to reduce crime, compared to both the base scenario and the other percentage increases, is to reassign patrolling time from 30% to 40% of the quadrants that are less insecure and less elastic (i.e., that are the least affected by police patrol time) and reassign it to the other 60%—70% of the quadrants such that they receive 20% more police patrol time. These results are depicted in Fig 3, which better displays the existing trade-off and the optimal reallocation strategy to reduce each type of crime.

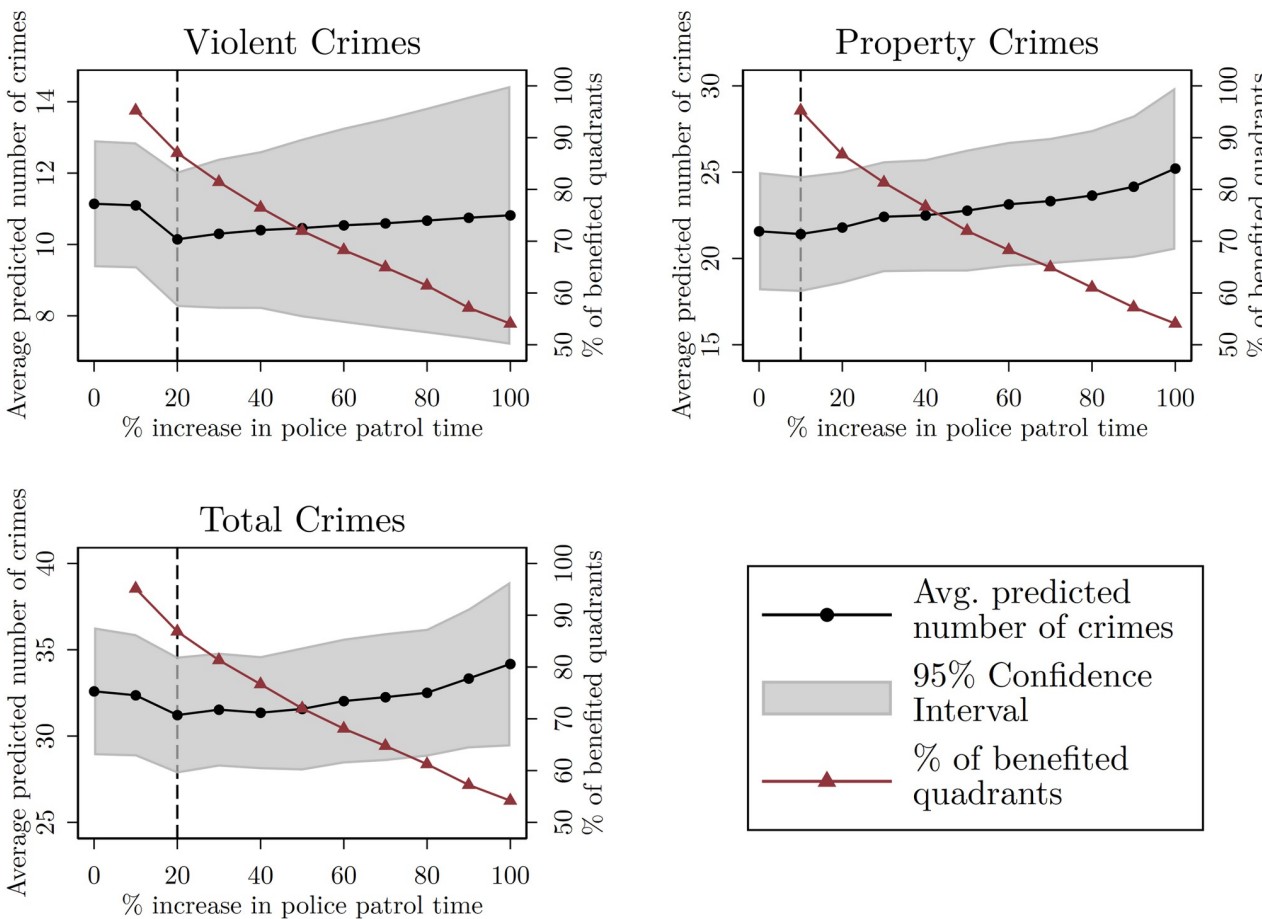

**Fig 3. Third policy scenario: Increment in police patrol time vs proportion of quadrants benefited (red line) and predicted average number of crimes per quadrant with 95% confidence intervals (black line).**

**Optimal policy.** We now estimate the optimal reassignment policy for this problem and we demonstrate its statistical significance. We also show that our model is consistent with [14] by showing that, conditional on our model, the random reassignment of police patrolling conducted in their study implies a modest reduction of crime in line with their claim that they can rule out statistically significant aggregate effects of more than 2%.

To solve for the optimal policy we solve the following optimization problem:

$$\min_{P_1,\dots,P_J} \sum_{j=1}^{J} NS_j(P_j, X_j, \xi_j; \alpha, \beta) \tag{13}$$

$$\sum_{j=1}^{J} P_j = TP \tag{14}$$

where *TP* is the observed total amount of time of police patrolling. This problem is equivalent

to:

$$\min_{P_1,\dots,P_J} \sum_{j=1}^{J} \exp(\alpha P_j + X_j \beta + \xi_j) \tag{15}$$

$$\sum_{j=1}^{J} P_j = TP \tag{16}$$

Intuitively, the optimization problem is equivalent to maximizing the outside option $S_0$.

Using our model, we construct an additional counterfactual scenario in which we estimate the amount of crime in case there had not been the intervention reported in [14] large scale experiment. For doing this, we identified the streets segments that where treated and set the police time to its original level (one half of the time observed for treated segments). Table 6 compares the results of three scenarios. The *Base scenario* corresponds to the prediction (fit) of

**Table 6. Base scenario, no intervention and optimal policy.**

| | 2*Observed | 2*Base scenario | 2*Optimal scenario | Predicted | | 2*No intervention scenario | | |
|---|---|---|---|---|---|---|---|---|
| | | | | Difference | | | Difference | |
| | | | | Point | Distribution | | Point | Distribution |
| *Panel A. Average crimes per cuadrant* | | | | | | | | |
| Violent crimes | 11.84 | 11.14 | 10.35 | -0.80*** | -0.58*** | 11.19 | 0.05 | 0.05*** |
| | (8.56) | (0.28) | (0.53) | [0.00] | [KS: 0.00; t: 0.00] | (0.30) | [0.65] | [KS: 0.00; t: 0.00] |
| Property crimes | 23.86 | 21.58 | 19.75 | -1.83*** | -1.36*** | 21.76 | 0.18 | 0.21*** |
| | (17.88) | (0.76) | (1.40) | [0.00] | [KS: 0.00; t: 0.00] | (0.86) | [0.65] | [KS: 0.00; t: 0.00] |
| Total crimes | 35.70 | 32.59 | 30.91 | -1.68*** | -1.40*** | 32.84 | 0.25 | 0.31*** |
| | (21.42) | (1.01) | (1.22) | [0.00] | [KS: 0.00; t: 0.00] | (1.19) | [0.68] | [KS: 0.00; t: 0.00] |
| *Panel B. Total crimes in the city* | | | | | | | | |
| Violent crimes | 12,435.00 | 11,698.57 | 10,862.81 | -835.77*** | -613.98*** | 11,749.84 | 51.26 | 56.56*** |
| | | (299.01) | (559.52) | [0.00] | [KS: 0.00; t: 0.00] | (316.67) | [0.65] | [KS: 0.00; t: 0.00] |
| Property crimes | 25,053.00 | 22,657.88 | 20,739.53 | -1,918.35*** | -1,432.26*** | 22,851.11 | 193.23 | 217.52*** |
| | | (794.91) | (1,470.22) | [0.00] | [KS: 0.00; t: 0.00] | (906.93) | [0.65] | [KS: 0.00; t: 0.00] |
| Total crimes | 37,488.00 | 34,222.57 | 32,459.56 | -1,763.01*** | -1,471.90*** | 34,483.95 | 261.37 | 321.37*** |
| | | (1,064.62) | (1,285.09) | [0.00] | [KS: 0.00; t: 0.00] | (1,244.58) | [0.68] | [KS: 0.00; t: 0.00] |

*Notes*:

*** p<0.01,

** p<0.05,

* p<0.1.

Standard deviation of the observed means and standard errors (SE) of the predictions in parentheses. Standard errors were obtained by a 1000-repetitions Bootstrap. p-values in squared brackets. KS: Kolmogorov-Smirnov test. t: t-test. Base scenario: crime predictions under the observed police patrol time. Optimal scenario: crime predictions under the optimal police patrol time distribution. No intervention scenario: crime predictions in the hypothetical case where no experiment was implemented. Difference corresponds to the rest between each of the las two scenarios and the base scenario. That is, Difference = Optimal—Base scenario, or Difference = No intervention—Base scenario. Two differences are presented, the point difference (Point), which corresponds to the difference between the predicted average (total) amount of crimes. The distribution difference (Distribution) correspond to the mean difference between the bootstrap distributions of predicted average (total) amount of crimes. For the Point difference, a one-sided p-value is reported. For the Distribution difference, p-values asociated to a two-sample Kolmogorov–Smirnov test for equality of distributions and a one-sided mean difference t-test. For the Optimal scenario, the Point p-value is computed as:

$p-value = \sum_{b=1}^{1000} 1\{\hat{C}^b \leq \hat{C}^*\}/1000$, where $\hat{C}^b$ is the prediction under the base scenario for bootstrap iteration $b$ and $\hat{C}^*$ is the average (sum) prediction under the optimal assignment. For the No intervention scenario, the p-value is computed as $p-value = \sum_{b=1}^{1000} 1\{\hat{C}^b \geq \hat{C}^{NI}\}/1000$, where $\hat{C}^{NI}$ is the average (sum) prediction under the hypothetical case of no experiment.

the model to the actual observed data. The *Optimal scenario* reports the results of solving the optimization problem above, and the *No intervention scenario* reports the results of the scenario in which we estimate the amount of crime in case there had not been an intervention. The table also reports average crimes and total crimes in the city for violent, property and total crimes. For example for violent crimes, the table shows that the optimal scenario has on average per quandrant of 0.80 less crimes than the Base scenario and it is a statistically significant with a 99% confidence level (bootstrapped *p*-values reported in square brackets under the point estimates column). Aggregate violent crimes are 835.77 less than in the Base scenario, and statistically significant with a 99% confidence level. Note also that in the No intervention scenario there is a small increase in all crimes relative to the Base scenario, but it is not statistically significant. This last result approximately reproduces the main result in [14].

Fig 4 shows the bootstrapped estimation of the uncertainty in the prediction of the average number of crimes per quadrant (Panel A) and total number of crimes (Panel B) in the base scenario. The vertical dash line shows the prediction of the model under [14] intervention and the red dashed line the prediction under the optimal policy. One sided *p*-values are reported in Table 6 in square brackets in the column of point estimates.

Fig 5 shows the bootstrapped estimation of the uncertainty for the three scenarios: No intervention policy with blue bars, base scenario ([14] intervention) with gray bars and the optimal patrol scenario with red bars. Clearly, the optimal scenario implies a considerable shift of the distributions to the left of the average number of crimes per quadrant (Panel A) and total number of crimes (Panel B). Table 6 shows that all of these distributions are statistically significant to a 99% confidence level. Table 6 (see Distribution columns) shows *p*-values associated to a two-sample Kolmogorov–Smirnov test for equality of distributions and a one-sided mean difference t-test. Although they are all significant differences the base scenario is marginally different to the no intervention scenario (still another confirmation of [14] results). However the optimal scenario shows a significant reduction in crime relative to the no intervention and the base scenarios.

## Discussion

In this paper we used a unique experimental data set [14] at the scale of a big urban center, Bogotá, the capital city of Colombia with more than seven million people. This randomized controlled trial was specially tailored towards the identification of the casual effect of police patrolling on crime. Using this data set, we proposed an estimation strategy based on a random utility model of crime location choice. In particular, our model, a conditional logit model with aggregate data, allows us to identify agents' fundamental parameters, such as their perceived utility of committing a crime. Once preferences were identified from observable data, we estimated the own- and cross-elasticities of crime to patrolling time and we were able to evaluate alternative patrolling strategies with the same total time cost. To the extent of our knowledge, both exercises are novel features in the literature studying the causal effect of police presence on crime levels in major cities in Latin America, and are very relevant from a public policy perspective. Our model gives results consistent with those reported by [14].

We found that police presence in a certain location has, on average, a negative impact on the utility of committing either a violent, a property, or any crime in that location: a one minute of police patrol time in location *j* reduces, on average, 0.006 units the utility of committing a violent crime and 0.008 units the utility of committing either property or any crime in that location. These effects are statistically significant with 95% confidence for violent crimes and 90% confidence for property and total crimes. Our most important results from a public policy perspective are the estimation of the own- and cross-elasticities of crime to patrolling time at

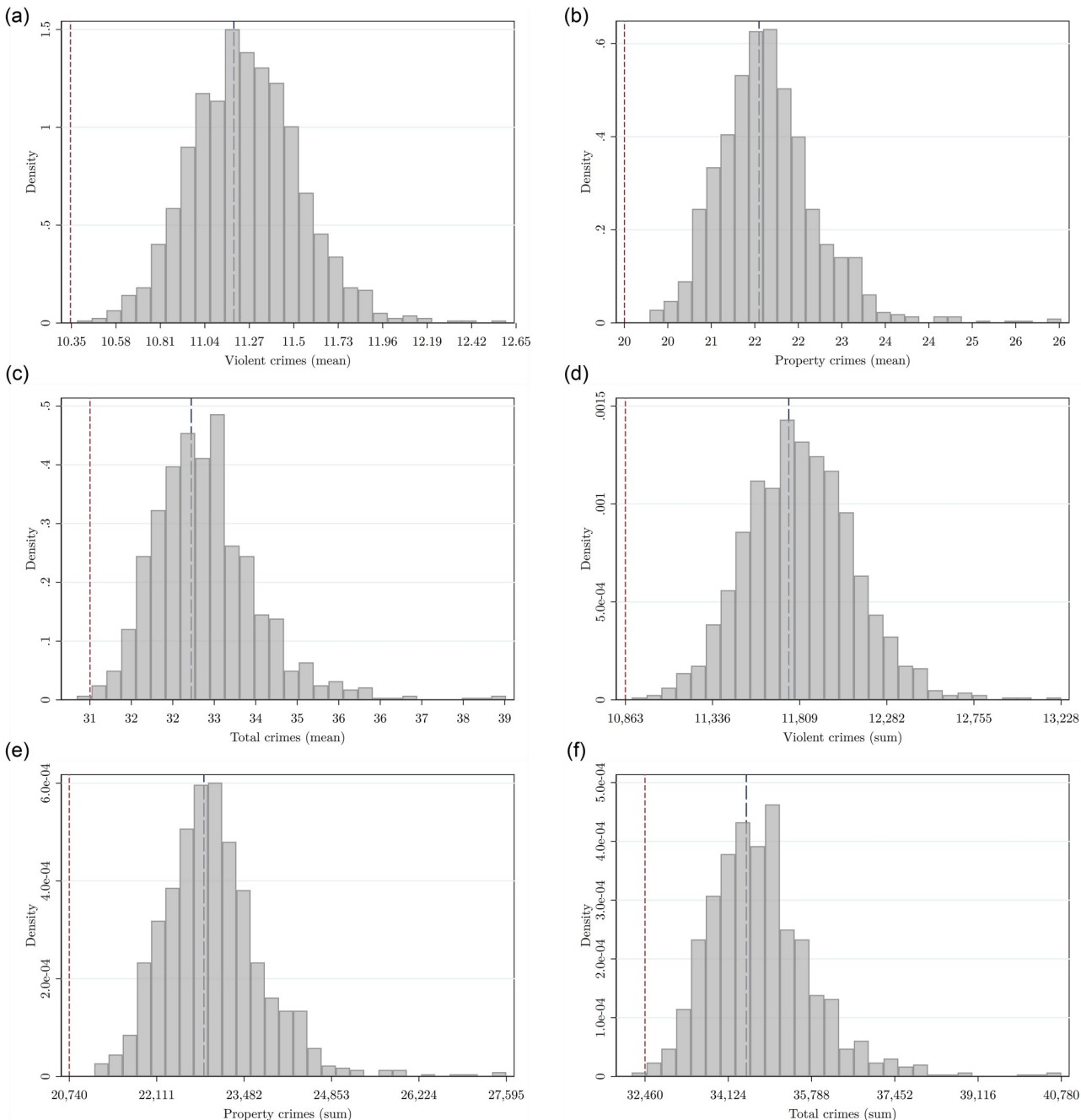

**Fig 4. Each figure displays the bootstrap distribution of the predicted average amount of crimes per quadrant (Panel A) and predicted sum of crimes in the city (Panel B) under the base scenario.** Vertical red (blue) dashed line displays the predicted mean and total number of crimes under the optimal (no intervention) police patrol scenario. (a) Violent crimes (b) Property crimes (c) Total crimes, (d) Violent crimes (e) Property crimes (f) Total crimes.

different locations, which captures the percentage change in crime that results from a 1% increase in the police patrol time that a location receives. To the extent of our knowledge, the estimation of these statistics in a properly identified structural model of crime location is new in the literature. For violent crimes, the elasticity is, on average, −0.26 and is statistically

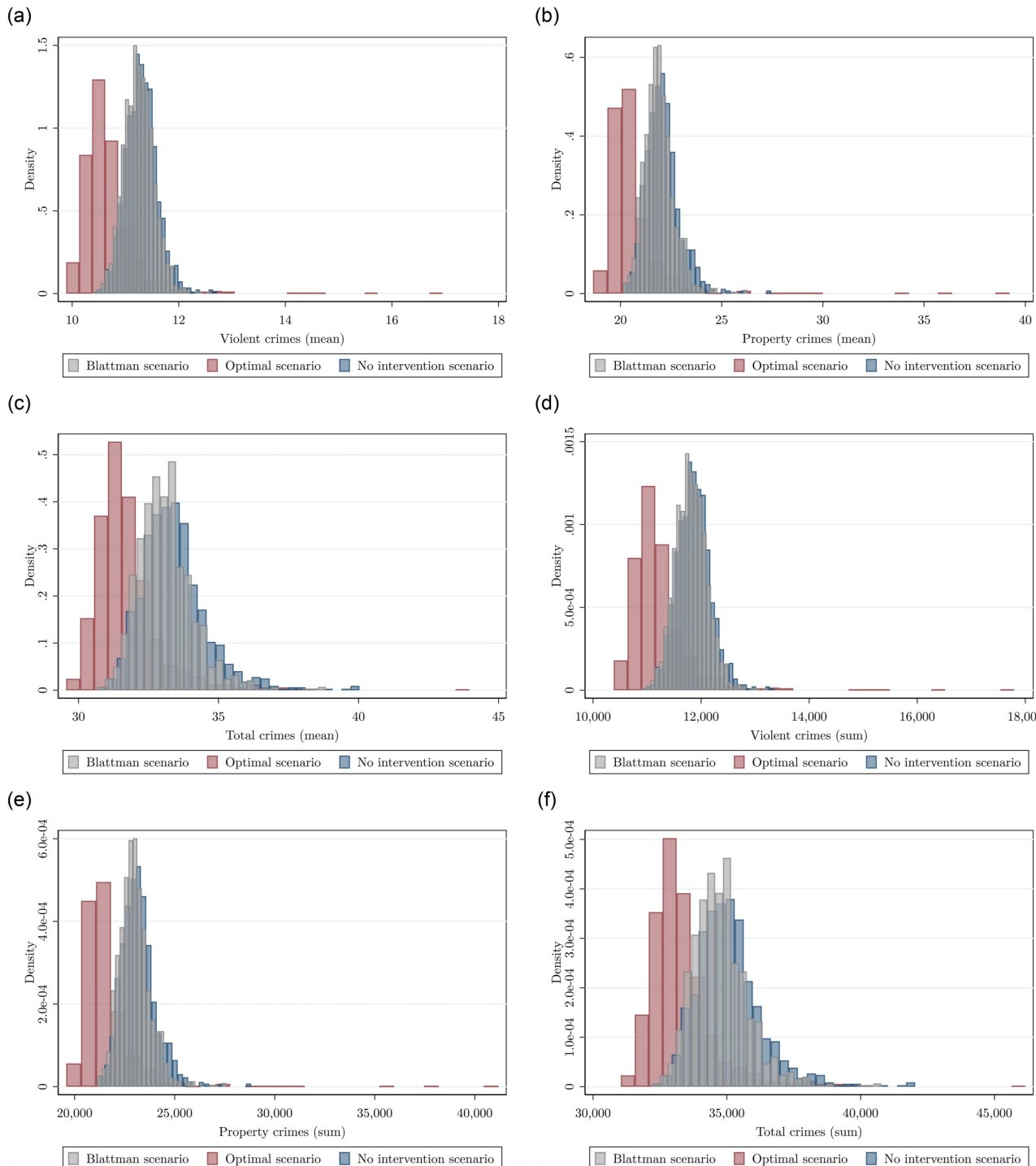

**Fig 5. Each figure displays the bootstrap distributions of the predicted average amount of crimes per quadrant (Panel A) and predicted sum of crimes in the city (Panel B) under the base scenario (gray bars), under the optimal police patrol scenario (red bars), and under the no-intervention scenario (blue bars).** (a) Violent crimes (b) Property crimes (c) Total crimes (d) Violent crimes (e) Property crimes (f) Total crimes.

significant at the 5% level. That is, a 10% increase in patrolling time in a quadrant reduces crime, on average across all quadrants, by 2.6% in that quadrant. The effect on property crime and all crimes is −0.38, 1.5 times larger. Moreover, our cross-police elasticity seems negligible suggesting there is no spillover effect. However, by construction, our model estimates an average cross-police elasticity across all quadrants, even those that are far away form the quadrants of interest. Therefore, it is natural to expect this average to be very low. Although small, when the effect is aggregated across the whole city, the effect may be non negligible. This stand in sharp contrast to the results of [14] that report a high spillover effect within street segments at less that 250 meters from treatment, suggesting that crime increases in the neighborhood of treated segments. In their paper, this spillover effect counterbalance the direct reduction in crime of police interventions and that is why the net effect that they report of the police allocation on crime is not statistically significant.

Taking advantage of our estimation of a structural model, we solve for the optimal reassignment of police time in the city. That is, instead of randomly choosing the road segments that receive double police patrols as in the original experiment [14], we assign this extra patrol time in a way that minimizes the expected level of crime in the 1,919 hot spots in Bogota. We show that it is possible to obtain an average reduction in violent crime of 7.09%, a reduction in property crimes of 8.48% and a reduction in total crimes of 5.15%. If we aggregate over all quadrants these implies a reduction of 835 in violent crimes, 1, 918 in property crimes and 1, 471 in total crimes during the period of study. This optimal reassignment policy uses the same amount of aggregate time as the current allocation policy.

Our results are in line with those of found by [14]. As shown in Table 6 and Fig 5 there is no statistical difference between the base scenario (i.e., [14] intervention) and the counterfactual of no intervention. This is due to the local and marginal nature of [14] intervention as compared to the optimal intervention. In the case of [14], the direct effect is counterbalanced by the spillover effect. In the optimal scenario, which we calculate using our structural model net effects are statistically significant and important from a public policy perspective.

Our conclusions are limited by our model and identifying assumptions. Our model is one of the simplest models in the discrete choice literature. It implies, for example, a constant cross-police elasticity from one quadrant to any other quadrant, no matter how far the other quadrant is. Clearly, the model and methodology would greatly benefit from introducing state of the art discrete choice models. In particular, models with stochastic coefficients as in [37], would allow for further treatment and agents heterogeneity and more realistic cross-police elasticities and counterfactual scenarios.

## Supporting information

**S1 Appendix. Police time reassignment strategies.**
(PDF)

**S1 Fig. Haussman-McFadden (1984) specification test of the Independence of Irrelevant Alternatives (IIA) assumption.** $\alpha$ estimated 1,018 times excluding in each iteration one different quadrant. Given that parameter estimates remain stable, IIA assumption seems to hold.
(PDF)

**S1 Table. OLS $\alpha$ estimates of the discrete spatial location choice model.** As a robustness check of our results, note that given the simultaneity that exists between crime and police presence, OLS estimates are downward biased. Therefore, if we estimate by OLS we would obtain lower bounds (in absolute value) of the real average impact of police presence on crime. As

can be seen form the table, results are indeed downward biased.
(PDF)

## Author Contributions

**Conceptualization:** Álvaro J. Riascos Villegas, Mateo Dulce Rubio.

**Formal analysis:** Douglas Newball-Ramírez, Álvaro J. Riascos Villegas, Mateo Dulce Rubio.

**Investigation:** Douglas Newball-Ramírez, Andrés Hoyos, Mateo Dulce Rubio.

**Methodology:** Álvaro J. Riascos Villegas, Mateo Dulce Rubio.

**Writing – original draft:** Douglas Newball-Ramírez, Álvaro J. Riascos Villegas.

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
