## [Decision Letter · Decision Letter 0]

29 Jun 2023

PONE-D-23-06563A Location Discrete Choice Model of Crime: Police Elasticity and Optimal DeploymentPLOS ONE

Dear Dr. Riascos Villegas,

Thank you for submitting your manuscript to PLOS ONE. After careful consideration, we feel that it has merit but does not fully meet PLOS ONE’s publication criteria as it currently stands. Therefore, we invite you to submit a revised version of the manuscript that addresses the points raised during the review process. Both reviewers agree that the study is interesting, and the methodology used sound. However, they both recommended improvements to the presentation, in particular the literature review, and more explicit statements of the research questions. Please submit your revised manuscript by Aug 12 2023 11:59PM. If you will need more time than this to complete your revisions, please reply to this message or contact the journal office at plosone@plos.org. Please include the following items when submitting your revised manuscript:A rebuttal letter that responds to each point raised by the academic editor and reviewer(s). You should upload this letter as a separate file labeled 'Response to Reviewers'.A marked-up copy of your manuscript that highlights changes made to the original version. You should upload this as a separate file labeled 'Revised Manuscript with Track Changes'.An unmarked version of your revised paper without tracked changes. You should upload this as a separate file labeled 'Manuscript'.

We look forward to receiving your revised manuscript.

Kind regards,

Siew Ann Cheong, Ph.D.

Academic Editor

PLOS ONE

Journal Requirements:

Reviewers' comments:

Reviewer's Responses to Questions

**Comments to the Author**

1. Is the manuscript technically sound, and do the data support the conclusions?

Reviewer #1: Yes

Reviewer #2: Yes

2. Has the statistical analysis been performed appropriately and rigorously? 

Reviewer #1: Yes

Reviewer #2: Yes

3. Have the authors made all data underlying the findings in their manuscript fully available?

Reviewer #1: Yes

Reviewer #2: Yes

4. Is the manuscript presented in an intelligible fashion and written in standard English?

Reviewer #1: Yes

Reviewer #2: Yes

5. Review Comments to the Author

Reviewer #1: The research is extremely important as it casts an eye on the study of crime prediction and the search for the type of strategy to be applied by police agencies. The literature specifies the application of random and targeted strategies, and this has to do with the type of criminal demand addressed. The present research presents a contribution for the discussion of the studied theme. However, I make some observations for the improvement of the present study:

1. The abstract could be improved, informing the reader of the limitations of the research, the practical and social implications of the research results, the gaps not addressed in the research, and indications for future research.

2. The authors did not present a specific section for literature review, in the introduction they presented the background of the problem, presenting several informative specific literatures that presented reports of research on hot spot. In this sense, I suggest the authors to consider the following texts, and that from them they can obtain other literatures that enrich the discussion: "Police and fear of crime in Distrito Federal"; "Predicting repeat offenders with machine learning: a casestudy of Beijing theives and burglars"; "Knowledge discovery in research on policing strategies: an overview of the past fifty years"; "Ranking policing strategies as a function of criminal complaints: application of the PROMETHEE II method in the Brazilian context"; "Identification of operational demand in law enforcement agencies: An application based on a probabilistic model of topics"; and "he spatial effect of police foot patrol on crime patterns: a localanalysis".

3. I suggest the authors to explicit in the introduction the problem question and the objectives that motivate the research.

4. In the introduction, the authors should introduce a short paragraph that summarizes the other sections of the article.

The author on page 8 cited as a footnote the following link to obtain the data: https://www.dane.gov.co/. However, when consulting this link I had difficulty in finding the source of the data used. I suggest that more detailed information should be provided, as it is important that other researchers may have access to this information, allowing the reproduction of this research.

6. In the conclusion, the authors indicate the implementation of random police strategies. The authors worked on the types of policing? do they indicate randomness for all strategies? I suggest the authors discuss this issue and reference with the authors studied.

Good review

Reviewer

Reviewer #2: Review: A Location Discrete Choice Model of Crime: Police Elasticity and Optimal Deployment.

The paper develops a discrete choice model of crime location with aggregate data in the city of Bogota (Colombia). Elasticities are estimated for each of the spatial locations and different police patrolling strategies are evaluated. The paper is interesting and provides a novel analysis for a region where there is not much reliable empirical data for this type of analysis.

That said, I think the paper can be improved in many ways. I am not going to get into the methodological heart of the paper but rather its presentation and conclusions.

1. Although the introduction is quite extensive, I believe that the paper would benefit from the incorporation of a more extensive literature review that would serve to bring the paper into the discussion of these types of models and their relationship to police deployment.

2. The literature review should also incorporate a section on Latin America. In the region there are already papers that analyze crime and police deployment from different methodological perspectives, including at the spatial level (geography of crime). It is necessary to insert this paper in that tradition.

3. Thirdly, it would be interesting to incorporate a discussion where the results are analyzed and then the conclusions where these results are connected with other literature, a future line of research is proposed to expand this analysis.

4. However, beyond these particular issues, the paper seems to move between two literatures that are not integrated in the text and neither at the level of analysis of results and conclusions: location of police resources and location of crime or concentration of crime. As it is, it is difficult to distinguish whether it is a contribution to the study of crime concentration patterns in large Latin American cities or whether it is intended as a contribution to the discussion on police distribution. In short: its results, which are valuable, are lost in part because they are not oriented to a particular debate.

5. There are no real conclusions in the paper either. A complete section should be included where the results are used to draw public policy conclusions on the level of concentration and distribution of crime or police deployment.

For the rest, I am grateful for the opportunity to have read this contribution, which can be a contribution to the discussion of crime in Latin America.

6. PLOS authors have the option to publish the peer review history of their article (what does this mean?). If published, this will include your full peer review and any attached files.

Reviewer #1: **Yes: **Prof. Dr. Marcio Basilio

Reviewer #2: No

---

## [Author Response · Author response to Decision Letter 0]

24 Aug 2023

PONE-D-23-06563

A Location Discrete Choice Model of Crime: Police Elasticity and Optimal Deployment

Plos One

Dear Plos One Editor,

We thank the reviewers for all the thoughtful comments and suggestions that helped improve our work's presentation and clarity. We appreciate the comments and hope our modifications and responses fulfill the requirements. All our responses appear in green, and modifications to the paper appear in the latex file in blue color to be easily identified. 

Reviewers' comments and responses:

Reviewer #1: The research is extremely important as it casts an eye on the study of crime prediction and the search for the type of strategy to be applied by police agencies. The literature specifies the application of random and targeted strategies, and this has to do with the type of criminal demand addressed. The present research presents a contribution for the discussion of the studied theme. However, I make some observations for the improvement of the present study:

1. The abstract could be improved, informing the reader of the limitations of the research, the practical and social implications of the research results, the gaps not addressed in the research, and indications for future research.

We have updated the abstract to better reflect the scope of the research:

Despite the common belief that police presence reduces crime, there is mixed evidence of such causal effects in major Latin America cities. In this work we identify the casual relationship between police presence and criminal events by using a large dataset of a randomized controlled police intervention in Bogotá D.C., Colombia. We use an Instrumental Variables approach to identify the causal effect of interest. Then we consistently estimate a Conditional Logit discrete choice model with aggregate data that allow us to identify agents' utilities for crime location using Two Stage Least Squares. The estimated parameters allow us to compute the police own and cross-elasticities of crime for each of the spatial locations and to evaluate different police patrolling strategies. The elasticity of crime to police presence is, on average across spatial locations, -0.26 for violent crime, -0.38 for property crime and -0.38 for total crime, all statistically significant. Estimates of cross-elasticities are close to zero; however, spillover effects are non-negligible. Counterfactual analysis of different police deployment strategies show, for an optimal allocating algorithm, an average reduction in violent crime of 7.09%, a reduction in property crimes of 8.48% and a reduction in total crimes of 5.15% at no additional cost. These results show the potential efficiency gains of using the model to deploy police resources in the city without increasing the total police time required.

2. The authors did not present a specific section for literature review, in the introduction they presented the background of the problem, presenting several informative specific literatures that presented reports of research on hot spot. In this sense, I suggest the authors to consider the following texts, and that from them they can obtain other literatures that enrich the discussion: "Police and fear of crime in Distrito Federal"; "Predicting repeat offenders with machine learning: a casestudy of Beijing theives and burglars"; "Knowledge discovery in research on policing strategies: an overview of the past fifty years"; "Ranking policing strategies as a function of criminal complaints: application of the PROMETHEE II method in the Brazilian context"; "Identification of operational demand in law enforcement agencies: An application based on a probabilistic model of topics"; and "he spatial effect of police foot patrol on crime patterns: a localanalysis".

We now have a separate literature review section.

We added “The spatial effect of police foot patrol on crime patterns: a local analysis” to the literature review since it analyzes the causal effect of police presence on crime. In detail, the authors study the spatial effect of police foot patrol in British Columbia, finding that the policy was effective in reducing crime when compared to the crime patterns in the same area before the program. This effect was concentrated in property crimes with no evidence of spatial crime displacement.

In addition, the literature review “Knowledge discovery in research on policing strategies: an overview of the past fifty years” was truly helpful and we added some articles analyzed there. In particular in the Topic 9 regarding experimental evidence of police strategies on crime levels.

3. I suggest the authors to explicit in the introduction the problem question and the objectives that motivate the research.

We have modified the introduction to answer this comment.

In detail, we are interested in studying the average causal effect on crime reduction of increasing police patrol time. Moreover, we are interested in estimating the own- and cross-elasticities of crime to patrol time at different locations, which captures the percentage change in crime, if any, that results from a 1% increase in the police patrol time that such location receives. This can also be interpreted as heterogeneous treatment effects for each spatial location. Finally, we use these elasticities to compare counterfactual policy scenarios in search of more efficient patrolling strategies.

4. In the introduction, the authors should introduce a short paragraph that summarizes the other sections of the article.

We have added this paragraph at the end of the introduction.

The rest of the paper is organized as follows. In the next subsection we summarize the relevant Related Literature and frame our work within the literature investigating the causal effect of police presence on crime events. The Materials and Methods section details the spatial discrete-choice model we use to model criminals' choices of where to commit crimes. Thereafter, we present our Empirical Strategy to estimate the parameters of interest along with our IV approach to identify the causal effect of police exposure on crime, while the Data section describes the experimental dataset used. In the Results section we analyze the estimations obtained regarding the causal effect of police presence on the level of crime, and the own- and cross- elasticities of such effect for each region of the city. In this section we use the estimated structural model to compare different counterfactual patrolling strategies. The Discussion section summarizes our findings and discusses limitations and next steps of our work.

5. The author on page 8 cited as a footnote the following link to obtain the data: https://www.dane.gov.co/. However, when consulting this link I had difficulty in finding the source of the data used. I suggest that more detailed information should be provided, as it is important that other researchers may have access to this information, allowing the reproduction of this research.

We have corrected these links: 

Data available at \\url{https://dane.gov.co/index.php/estadisticas-por-tema/mercado-laboral/empleo-y-desempleo/mercado-laboral-historicos#marco-2005} and \\url{https://www.dane.gov.co/index.php/estadisticas-por-tema/demografia-y-poblacion/proyecciones-de-poblacion}

6. In the conclusion, the authors indicate the implementation of random police strategies. The authors worked on the types of policing? do they indicate randomness for all strategies? I suggest the authors discuss this issue and reference with the authors studied.

Thanks for giving us an opportunity to clarify this issue. The randomized control trial conducted by Blattman et.al. 2021 in Bogota was an experimental design that we describe briefly in the literature review section:

The authors randomly assigned 1,919 streets to an 8-month treatment of doubled police patrols, greater municipal services, both, or neither. They studied the direct and spillover effects of such targeted state services. 

Reviewer #2: Review: A Location Discrete Choice Model of Crime: Police Elasticity and Optimal Deployment.

The paper develops a discrete choice model of crime location with aggregate data in the city of Bogota (Colombia). Elasticities are estimated for each of the spatial locations and different police patrolling strategies are evaluated. The paper is interesting and provides a novel analysis for a region where there is not much reliable empirical data for this type of analysis.

That said, I think the paper can be improved in many ways. I am not going to get into the methodological heart of the paper but rather its presentation and conclusions.

1. Although the introduction is quite extensive, I believe that the paper would benefit from the incorporation of a more extensive literature review that would serve to bring the paper into the discussion of these types of models and their relationship to police deployment.

We have added a comprehensive literature review section separated from the introduction. In this subsection we frame our work within the literature investigating the causal effect of police presence on crime events

2. The literature review should also incorporate a section on Latin America. In the region there are already papers that analyze crime and police deployment from different methodological perspectives, including at the spatial level (geography of crime). It is necessary to insert this paper in that tradition.

We have added a paragraph summarizing recent works studying the causal effect of proactive police strategy in major Latin America cities.

3. Thirdly, it would be interesting to incorporate a discussion where the results are analyzed and then the conclusions where these results are connected with other literature, a future line of research is proposed to expand this analysis.

In the discussion section we have directly compared our results to those of Blattman et.al 2021. This paper is directly related to ours and we have used their data to identify our structural model. As explained in the paper and discussion section our results are consistent with their results but, since we have estimated a structural model, we can analyze more police allocating scenarios. 

We have also added one particular line for future research that we believe is very promising. 

4. However, beyond these particular issues, the paper seems to move between two literatures that are not integrated in the text and neither at the level of analysis of results and conclusions: location of police resources and location of crime or concentration of crime. As it is, it is difficult to distinguish whether it is a contribution to the study of crime concentration patterns in large Latin American cities or whether it is intended as a contribution to the discussion on police distribution. In short: its results, which are valuable, are lost in part because they are not oriented to a particular debate.

Thanks for giving us an opportunity to clarify this issue. The randomized control trial conducted by Blattman et al. (2021) in Bogotá was an experimental design that we describe briefly in the literature review section: The authors labeled 1,919 streets as hotspots according to their pre-experiment crime level between 2012 and 2015. These 1,919 streets were randomly selected to an 8-month treatment of dual police patrol, increased municipal services, both, or neither. In particular, 756 out of the 1,919 hotspots were randomly assigned to double police patrol which allows them to study the direct and indirect effects of such targeted state services. In our case, instead of randomly choosing the street segments that receive double police patrolling as in the original experiment, we assign these extra police time in a way that minimizes the expected level of crime in the 1,919 hot spots of Bogotá.

5. There are no real conclusions in the paper either. A complete section should be included where the results are used to draw public policy conclusions on the level of concentration and distribution of crime or police deployment.

In the new version of the paper, following Plos One publication template, we have concentrated all results and analysis in the results section. For clarity, this results section is divided in two subsections. In one section we report and analyze the results of the estimation of the structural model and in the second subsection we consider several counterfactual policy scenarios.

In the discussion section we have highlighted the main results of the paper in terms of causal effects of police presence and the potential benefits of an optimal use of police time allocation.

---

## [Decision Letter · Decision Letter 1]

25 Oct 2023

A Location Discrete Choice Model of Crime: Police Elasticity and Optimal Deployment

PONE-D-23-06563R1

Dear Dr. Riascos Villegas,

We’re pleased to inform you that your manuscript has been judged scientifically suitable for publication and will be formally accepted for publication once it meets all outstanding technical requirements.

Kind regards,

Hanna Landenmark

Staff Editor

PLOS ONE

on behalf of 

Siew Ann Cheong, Ph.D.

Academic Editor

PLOS ONE

Additional Editor Comments (optional):

Comments from PLOS Editorial Office: We note that one or more reviewers has recommended that you cite specific previously published works. As always, we recommend that you please review and evaluate the requested works to determine whether they are relevant and should be cited. It is not a requirement to cite these works. We appreciate your attention to this request.

Comments from Staff Editor Hanna Landenmark (hlandenmark@plos.org): Please ensure that the code is shared when you resubmit the final proofs, and update the Data availability statement to declare where the code can be found. Please also include a link to the code within the manuscript text, e.g. in the Methods section.

Reviewers' comments:

Reviewer's Responses to Questions

**Comments to the Author**

1. If the authors have adequately addressed your comments raised in a previous round of review and you feel that this manuscript is now acceptable for publication, you may indicate that here to bypass the “Comments to the Author” section, enter your conflict of interest statement in the “Confidential to Editor” section, and submit your "Accept" recommendation.

Reviewer #1: All comments have been addressed

Reviewer #2: All comments have been addressed

2. Is the manuscript technically sound, and do the data support the conclusions?

Reviewer #1: Yes

Reviewer #2: Yes

3. Has the statistical analysis been performed appropriately and rigorously? 

Reviewer #1: N/A

Reviewer #2: Yes

4. Have the authors made all data underlying the findings in their manuscript fully available?

Reviewer #1: Yes

Reviewer #2: Yes

5. Is the manuscript presented in an intelligible fashion and written in standard English?

Reviewer #1: Yes

Reviewer #2: Yes

6. Review Comments to the Author

Reviewer #1: Dear Authors

I would like to congratulate you on your extensive revision of the text. I can see that you have answered almost all of the reviewers' questions. The text is much improved. However, I would like to suggest a few changes that the authors say they have made, but which I have not seen in the main text. I believe there must have been some mistake when compiling the new version. I would therefore ask you to answer the following questions:

1) The authors state that they have inserted the following paper: "Knowledge discovery in research on policing strategies: an overview of the past fifty years", Journal of Modelling in Management, Vol. 17 No. 4, pp. 1372-1409. " ext-link-type="uri" xlink:type="simple">https://doi.org/10.1108/JM2-10-2020-0268" However, I did not notice it in the references. I suggest you include it.

2) As for the other suggestions for bibliography that I suggested in the first round of reviews, I would like the authors to comment on their lack of suitability for the topic: https://doi.org/10.1108/JM2-05-2020-0122;
https://doi.org/10.1108/JM2-01-2018-0001

Good review.

Reviewer #2: (No Response)

7. PLOS authors have the option to publish the peer review history of their article (what does this mean?). If published, this will include your full peer review and any attached files.

Reviewer #1: **Yes: **Dr. Marcio Basilio

Reviewer #2: **Yes: **Gustavo Fondevila

---

## [Editor Report · Acceptance letter]

28 Dec 2023

PONE-D-23-06563R1 

PLOS ONE

Dear Dr. Riascos Villegas, 

I'm pleased to inform you that your manuscript has been deemed suitable for publication in PLOS ONE. Congratulations! Your manuscript is now being handed over to our production team.

Kind regards, 

on behalf of

Dr. Siew Ann Cheong 

Academic Editor

PLOS ONE